# Learning to Express in Knowledge-Grounded Conversation

## Abstract

Grounding dialogue generation by extra knowledge has shown great potentials towards building a system capable of replying with knowledgeable and engaging responses. Existing studies focus on how to synthesize a response with proper knowledge, yet neglect that the same knowledge could be expressed differently by speakers even under the same context. In this work, we mainly consider two aspects of knowledge expression, namely the structure of the response and style of the content in each part. We therefore introduce two sequential latent variables to represent the structure and the content style respectively. We propose a segmentation-based generation model and optimize the model by a variational approach to discover the underlying pattern of knowledge expression in a response. Evaluation results on two benchmarks indicate that our model can learn the structure style defined by a few examples and generate responses in desired content style.

## 1 Introduction

Human-machine conversation is a long-standing goal of artificial intelligence (AI). In the past few years, with advances in deep learning [28, 5, 31] and availability of huge amount of human conversations on social media [1], building an open domain dialogue system with data-driven approaches has attracted increasing attention from the community of AI and NLP. By synthesizing a response with text generation techniques [32], current natural models are able to naturally reply to user prompts. Despite the impressive progress, existing generation models are notorious for replying with generic and bland responses, resulting in meaningless and boring conversations [13]. Such deficiency is particularly severe when human participants attempt to dive into specific topics in conversation [3].

To bridge the gap, some researchers resort to ground dialogue generation by extra knowledge such as unstructured documents [49, 3]. By this means, the documents (e.g., wiki articles) serve as content sources and make a dialogue system knowledgeable regarding various concepts in a discussion. However, existing studies focus on how to synthesize a response with proper knowledge [3, 11, 45], but pay little attention to the fact that the same knowledge could be expressed differently even under the same context. These models usually employ a regular decoder to generate the response in an auto-regressive manner given the contextual representations of knowledge and dialogue context, which makes the generation process less explainable and controllable.

In general, we break down the expression style of a response into two components: the structure of the response and the style of the content in each part. First, the knowledge expression in response varies in structure, including but not limited to the position and the length of knowledge expression. As the example shown in Table 1, knowledge-related phrases and clauses could be long, like "And I'd give credit to three different voice actors for anna.", or short, like "74 in Metacritics". Besides, they may appear at the beginning of the sentence, or at the end. For the sake of description, we decompose a response into a sequence of non-overlapping segments, each is either related to certain background

Table 1: A case from CMU_DoG. Given the same knowledge and context, the last two turns in left and right conversations exhibit positive and negative sentiments, respectively. Each utterance can be decomposed into knowledge-related and knowledge-irrelevant segments.

| Knowledge |
| --- |
| • MovieName: Frozen
• Year: 2013
• Rating: Rotten Tomatoes: 89% , Metacritics: 74/100, CinemaScore: A+
• Genre: Comedy, Adventure, Animation
• Director: Chris Buck, Jennifer Lee
• Cast: Kristen Bell as Anna, the 18-year-old Princess of Arendelle and Elsa's younger sister, Livvy Stubenrauch as 5-year-old Anna, Katie Lopez as 5-year-old Anna (singing), Agatha Lee Monn as 9-year-old Anna ...
• ... |

| Conversations | |
| --- | --- |
| User1: I was really surprised that disney chose Kristen Bell to be the voice of Anna in Frozen
User2: Yes, I didn't imagine it'd be her!
User2: What do you think about the rating?
User1: 74 in Metacritics. I believe it deserves, indeed.
User1: And I'd give credit to three different voice actors for anna. I'm really impressed. What about you?
... | User1: I was really surprised that disney chose Kristen Bell to be the voice of Anna in Frozen
User2: Yes, I didn't imagine it'd be her!
User2: What do you think about the rating?
User1: The rating is 74 in Metacritics. Let me say, high enough for a Disney move
User1: And I do think it was overkill to use three different voice actors for anna. Do you agree ?
... |

knowledge and diverse in content style, or almost irrelevant to the knowledge but simply playing the role of stitching the context and carrying on the conversation. We therefore define the structure style as the distribution and number of two kinds of segments. Structure style itself is far from dominant in the sentence expression, since different speakers could convey converse attitude even the context and the knowledge are exactly the same, as shown in Table 1. So it is necessary to introduce the content style as the expression fashion within each knowledge-related segment. We further introduce two latent variables to facilitate end-to-end training, one for predicting the start and end positions of a segment, the other for deciding the category of each segment. Since the human annotations for sentence segmentation are absent and enumerating over all possibilities to maximize the likelihood of the response is time-consuming, we propose a variational framework for segmentation-based generation and induce an evidence lower bound of the likelihood.

Formally, our model is on the basis of encoder-decoder architecture. The encoder is to obtain the contextual representation of conversational context and knowledge in a regular way. The decoder consists of three types of modules: (1) a context module, for response only based on context without knowledge; (2) a plain-knowledge module, for response referring knowledges but without particular style; and (3) one or more stylized-knowledge module, for response referring knowledges and with a specific style. The context module is the only module not relying on knowledge, but simply paying attention to contextual information. Compared with plain-knowledge module, stylized-knowledge module has unique adapters, which is their primary discrepancy. When decoding, the decoder first predicts the segmentation of the response and then makes a choice in three kinds of modules to generate a single segment. Both the segmentation and the module selection are instructed under sequential latent variables.

We train our model on the Reddit Corpus published by [15] and evaluate our model on two benchmarks of knowledge-grounded conversation: Wizard of Wikipedia(Wizard) [3] and CMU Document Grounded Conversation(CMU_DoG) [49]. Evaluation results indicate that our model can significantly outperform state-of-the-art methods in the zero-resource setting (i.e., only trained on the Reddit Corpus). In addition, the performance of our model improves significantly on Wizard and CMU_DoG with the presence of only $10\%$ training data and the segment distributions after fine-tuning are consistent with our prior knowledge about the two datasets, indicating that our model can learn the structure style with little cost. Finally, our model outperforms previous state-of-the-art models on the accuracy of performing sentiment classification using generated responses. It is worth noting that our model achieves $10\%+$ accuracy improvement on Wizard Seen, $12\%+$ accuracy improvement on Wizard Unseen, and $12\%+$ accuracy improvement on CMU_DoG than the present state-of-the-art model, which indicates that the model can be controlled to express knowledge with the desired content style.

Contributions in this work are three-fold: (1) exploration the knowledge expression in knowledge-grounded conversation; (2) proposal of a variational segmentation-based generation model to discover the underlying expression style in a response; (3) empirical verification of the effectiveness of the proposed model on two benchmarks of knowledge-grounded conversation.

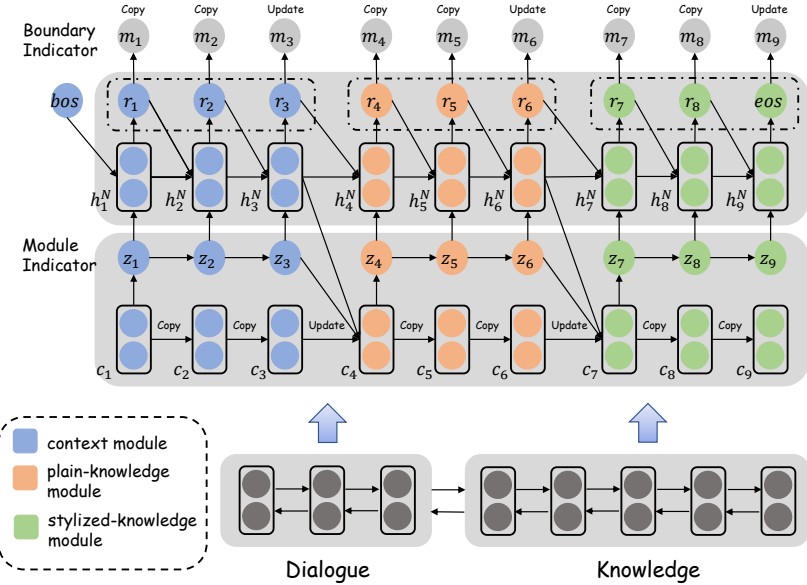

Figure 1: Architecture of the proposed model.

## 2 Related Work

Early research on end-to-end open-domain dialogue generation is inspired by the successful application of neural networks on machine translation [20, 24, 32]. On the vanilla encoder-decoder architecture, various extensions have been made to model the structure of dialogue contexts [22, 23, 37, 39]; to improve diversity of responses [13, 36, 43, 29]; to control attributes of responses [38, 46, 40, 34, 21]; and to bias responses to some specific personas [14, 41]. Recently, grounding dialogue generation by extra knowledge has seemed promising to bridge the gap between conversation with existing systems and conversation with humans, and the knowledge could be obtained from knowledge graphs [48, 18, 30], retrieved from unstructured documents [3, 16, 44, 11, 45, 15], or extracted from visual background [19, 26, 9]. In this work, we study document-grounded dialogue generation. Rather than selecting knowledge relevant to dialogue context and directly exploiting pre-trained language models to generate the response, we focus on expressing knowledge in this task.

The idea of sequence modeling via segmentation [33] has attracted widespread attention in several natural language processing tasks. In the field of text segmentation, [33] propose a probabilistic model for sequence modeling via their segmentation and a "Sleep-WAke Network"(SWAN) method. In machine translation, [8] propose a neural phrase-based machine translation system that models phrase structures in the target language using SWAN. In data-to-text generation, [35] develop a neural, template-like generation model based on an HSMM decoder, which can be learned tractably by backpropagating through a dynamic program; to tackle the problem of weak Markov assumption for the segment transition probability, [25] propose to explicitly segment target text into fragments and align them with their data correspondences, and jointly learn the segmentation and correspondence via dynamic programming. Though quite a few methods have been proposed to reduce the computational complexity [33, 25], using dynamic programming to calculate likelihood is still expensive. This work introduces two sequential latent variables to model the knowledge expression and proposes a variational segmentation-based generation framework, which enjoys less computation cost.

## 3 Approach

### 3.1 Problem Formalization and Motivation

Suppose that we have a dataset $\mathcal{D} = \{(U_i, K_i, R_i)\}_{i=1}^N$, where $\forall i \in \{1, \ldots, N\}$, $K_i$ serves as background knowledge of the dialogue $(U_i, R_i)$ with $K_{i,j}$ the $j$-th sentence, $U_i$ is the context of the dialogue with $U_{i,j}$ the $j$-th utterance, and $R_i$ is the response. To bias the expression to a specific structure style, we further assume that there are a few examples $\mathcal{D}_{sty} = \{(U_i, K_i, R_i)\}_{i=1}^M$ provided by users depicting the required style for knowledge expression. Note that we have $N \gg M$, since corpus in a specific expression style is rare and difficult to acquire. The goal is to learn a generation model

$p_\theta(R|U,K)$ ($\theta$ denotes the parameters of the model) from $\mathcal{D}$, to generate a response $R$ following $p_\theta(R|U,K)$ given a new dialogue context $U$ and the associated knowledge $K$. Besides, one can either (1) bias the structure style of $P_\theta(R|U,K)$ to $\mathcal{D}_{sty}$ with little cost; or (2) switch the content style of knowledge expression in $R$.

As mentioned above, the response can be decomposed into a sequence of segments, each is other knowledge-related, various in expression, or knowledge-irrelevant. Therefore manipulating the expression style of a response could be split into two subproblems. One is to control the structure style, in other word, distribution and number of two kinds of segments. The other subproblem is the content style, or generating every knowledge-related segment in desired content style, such as positive style or negative style or other customized styles defined by users. To solve the two subproblems, we propose a segmentation-based generation model, which could automatically detect and predict the segmental structure of the response and then generate segments of a response one by one. Each segment is either knowledge-irrelevant or knowledge-related, and knowledge-related segments could be expressed in arbitrary style, defined and manipulated by users. Both the segmentation and the choice are modeled by a latent variable, so as to facilitate end-to-end training. Furthermore, to guarantee the efficiency and practicality of our model, we propose a variational approach to optimize the evidence lower bound (ELBO) of the likelihood of response to circumvent directly marginalizing over all possible combinations of segmentation and action choice, which is time-consuming in both training and test stages.

## 3.2 Model Architecture

Figure 1 gives an overview of the proposed model, which is based on the encoder-decoder architecture. The encoder generates the contextual representations of the dialogue and knowledge, while the decoder generates the segments one after another. $\mathbf{h}_t^N$ encodes the dialogue context up to timestep $t-1$ with $N$ denoting the number of decoder layers. Given $R = (r_1, \cdots, r_t, \cdots, r_{l_r})$ with $r_t$ referring the $t$-th token of $R$ whose length is supposed to be $l_r$, the variable $Z = \{z_t\}_{t=1}^{l_r}$ is utilized to control the choice of module of each segment(**Module Indicator**), and its historical information is encoded by $\{\mathbf{c}_t\}_{t=0}^{l_r}$. $M = \{m_t\}_{t=1}^{l_r}$ is a sequence of binary variables and used to determine the boundary of each segment(**Boundary Indicator**). Specifically, $m_t = 1$ indicates that the current segment is already completed and a new segment should be created at the next timestep. Otherwise $m_t = 0$ and the current segment remains unfinished. The generative process is disassembled into two steps: (1) determine the type of a new segment based on previously generated text and previous segment types; (2) generate within the current segment until the binary variable $m_t = 1$.

**Context and Knowledge Encoding.** We exploit BART[12] as the backbone of our architecture, which is pre-trained using a variety of denoising objectives and achieves state-of-the-art results on a range of text generation tasks. Given the dialogue context $U = (U_1, \cdots, U_n)$, we simply concatenate them as $(u_1, \cdots, u_{l_u})$. Similarly, we concatenate the associated knowledge $K = (K_1, \cdots, K_m)$ as $(k_1, \cdots, k_{l_k})$. $l_u$ and $l_k$ are the length of dialogue context and background knowledge respectively. The input of the encoder is then defined as:

$$I = [\text{BOS}]k_1 \ldots k_{l_k}[\text{EOS}]u_1 \ldots u_{l_u}[\text{EOS}]. \tag{1}$$

The input $I$ then passes through the stacked self-attention layers and results in a knowledge-aware context representation $\mathbf{C}$, and a context-aware knowledge representation $\mathbf{K}$. Specifically, the context-aware knowledge representation is defined as $\mathbf{K} = [\mathbf{h}_1^{enc}, \cdots, \mathbf{h}_{l_k+1}^{enc}]$ where $\mathbf{h}_t^{enc}$ is the last layer of BART encoder at time $t$. Similarly, the knowledge-aware context representation is defined as $\mathbf{C} = [\mathbf{h}_{l_k+2}^{enc}, \cdots, \mathbf{h}_{l_k+l_u+2}^{enc}]$.

**Prior of Module Indicator.** We use the sequential discrete latent variable $Z = \{z_t\}_{t=1}^{l_r}$ to decide which module to invoke at each timestep. The transition of $z_t$ occurs only when a segment is completed, which is decided by the binary boundary variable $M$. The prior quantifies the distribution of $z_t$ before we observe the segment, and it is reasonable to assume that the prior of $z_t$ depends on previous module choices $z_{<t}$ and previously generated text. As a result, the transition of $Z$ is implemented as follows:

$$p_{\theta_z}(z_t|r_{<t}, z_{<t}, m_{t-1}) = m_{t-1} \cdot \tilde{p}(z_t|\mathbf{c}_t) + (1 - m_{t-1}) \cdot \delta(z_t = z_{t-1}), \tag{2}$$

where $\mathbf{c}_t$ encodes all previous latent state $z_{<t}$ and generated text $r_{<t}$ as follows:

$$\mathbf{c}_t = m_{t-1} \cdot f_{z-\mathrm{rnn}}(\tilde{\mathbf{z}}_{t-1}, \mathbf{c}_{t-1}) + (1 - m_{t-1}) \cdot \mathbf{c}_{t-1}. \tag{3}$$

$\tilde{\mathbf{z}}_{t-1} = [\mathbf{e}_{t-1}; \mathbf{h}_{t-1}^{N,dec}]$ with $\mathbf{e}_{t-1}$ the embedding of $z_{t-1}$ and $\mathbf{h}_{t-1}^{N,dec}$ the representation of last generated token. Specifically, $m_{t-1} = 0$ means that the next timestep $t$ is still in the same segment as the previous timestep $t-1$ and thus the latent variable $z_t$ should not be updated. Otherwise, it means that current segment is completed and $z_t$ is updated with the transition function $\tilde{p}(z_t|\mathbf{c}_t)$. Because we only have $N_{sty} + 2$ options when choosing a module, where $N_{sty}$ is the number of different user-defined styles, in addition with 2 default styles, so in this model, the latent variable $z_t$ ranges in natural integer to denote corresponding style type. Specifically, $z_t = 0$ denotes choosing the context expression module to generate a knowledge-irrelevant segment; $z_t = 1$ tells the model to choose the knowledge expression module without specially customized style; we leave the $z_t \geq 2$ to be user-defined so as to select the knowledge expression module combined with customized style. The transition function $\tilde{p}(z_t|\mathbf{c}_t)$ is then implemented as a multinomial distribution parameterized by $\mathrm{Softmax}(f_{z-\mathrm{mlp}}(\mathbf{c}_t))$.

**Prior of Boundary Indicator.** The boundary indicator $M = \{m_t\}_{t=1}^{l_r}$ depicts the segmental structure of the response, with $m_t = 1$ indicates that a new segment will start at time $t + 1$. Presumably, the prior of $m_t$ could be inferred from $r_{\leq t}$ and $z_t$. We model the distribution $p_{\theta_m}(m_t|r_{\leq t}, z_t)$ by a Bernoulli distribution parameterized by $\sigma(f_{m-\mathrm{mlp}}([\mathbf{e}_{t-1}; \mathbf{h}_{z_{t-1},t-1}^{N,dec}]))$, where $\sigma$ denotes the sigmoid function and $f_{m-\mathrm{mlp}}$ is a multi-layer perceptron network.

**Stylized Generation** As mentioned above, the generation process involves scheduling different modules according to $z_t$. Here we give a systematic description of the generation process. The decoder accepts the token generated last timestep $r_{t-1}$ as input, performs transformation in $N$ decoder layers, finally obtains a dense representation.

We use $\mathbf{h}_t^l$ to denote the hidden state after the $l$-th layer at timestep $t$, which is a shorthand for $\mathbf{h}_t^{l,dec}$ for brevity. Specially, $\mathbf{h}_t^0$ is the output of the embedding layer. When $z_t = 0$, it implies that knowledge encoding is unnecessary for current segment so $\mathbf{h}_t^l$ is defined as:

$$\mathbf{h}_t^l = \mathrm{DecoderLayer}(\mathbf{h}_t^{l-1}, \mathbf{H}_{t-1}^{l-1}, \mathbf{C}), \tag{4}$$

where $\mathbf{H}_{t-1}^{l-1} = [\mathbf{h}_1^l, \cdots, \mathbf{h}_{t-1}^l]$ is a sequence of decoder hidden states in previous timestep, and $\mathbf{C}$ is the context representation mentioned above. The implementation of $\mathrm{DecoderLayer}(\cdot, \cdot, \cdot)$ is identical to the vanilla Transformer [31] where $\mathbf{h}_t^{l-1}$ first plays self-attention on $\mathbf{H}_{t-1}^{l-1}$ then performs cross-attention on $\mathbf{C}$. The probability $p(r_t|r_{<t}, z_t = 0)$ is defined as a multinomial distribution parameterized by $\mathrm{Softmax}(f_{r-\mathrm{mlp}}(\mathbf{h}_t^N))$, where $\mathbf{h}_t^N$ encodes the generated tokens up to timestep $t-1$. When $z_t = 1$, the implementation of decoder layer is analogous to the $z_t = 0$ case except that we replace $\mathbf{C}$ with $\mathbf{K}$, since knowledge is needed:

$$\mathbf{h}_t^l = \mathrm{DecoderLayer}(\mathbf{h}_t^{l-1}, \mathbf{H}_{t-1}^{l-1}, \mathbf{K}). \tag{5}$$

To generate a segment with a particular customized style when $z_t \geq 2$, we introduce some adapters [7] to bias the generation. Specifically, the hidden state $\mathbf{h}_t^l$ is defined as:

$$\mathbf{h}_t^l = \mathrm{DecoderLayer}_{adp}(\mathbf{h}_t^{l-1}, \mathbf{H}_{t-1}^{l-1}, \mathbf{K}), \tag{6}$$

where $\mathrm{DecoderLayer}_{adp}(\cdot, \cdot, \cdot)$ denotes the transformer decoder layer with adapters inserted. Note that we need to introduce a separate set of adapters for each style. To make the style fine-grained and adjustable, each style has a unique set of adapters. Different styles have no adapter in common. In addition, our model has the ability to learn to express in any type of style, as long as a discriminator for the desired style is provided.

## 3.3 Learning Details

We introduce auxiliary distributions $q_{\phi_m}(M|R) = \prod_{t=1}^{l_r} q_{\phi_m}(m_t|R)$ and $q_{\phi_z}(Z|M, R) = \prod_{t=1}^{l_r} q_{\phi_z}(z_t|M, R)$, which serves as an approximation to the intractable posterior of the bound-

ary indicator $M$ and the module indicator $Z$. We then apply variational approximation which gives the following evidence lower bound objective [1](ELBO)[6]:

$$\log p_\theta(R|U, K) \geq \mathbb{E}_{q_{\phi_m}(M|R)} \left( \mathbb{E}_{q_{\phi_z}(Z|M,R)} \sum_{t=1}^{l_r} \log p_\theta(r_t|r_{<t}, z_t) \right.$$
$$\left. - \sum_{t=1}^{l_r} m_{t-1} \cdot D_{\mathrm{KL}}\big(q_{\phi_z}(z_t|M, R) \| p_{\theta_z}(z_t)\big) \right) - \sum_{t=1}^{l_r} D_{\mathrm{KL}}\big(q_{\phi_m}(m_t|R) \| p_{\theta_m}(m_t)\big), \tag{7}$$

where $p_{\theta_z}(z_t)$ and $p_{\theta_m}(m_t)$ stand for $p_{\theta_z}(z_t|r_{<t}, z_{<t}, m_{t-1})$ and $p_{\theta_m}(m_t|r_{\leq t}, z_t)$ respectively, and $D_{\mathrm{KL}}(\cdot\|\cdot))$ refers to Kullback–Leibler divergence. Detailed derivations are presented in supplementary material.

Base on the intuition that the response provides hints about the segmentation, we construct the posterior distribution $q_{\phi_m}(m_t|R)$ as a Bernoulli distribution parameterized by $\sigma(f'_{m-\mathrm{mlp}}(\psi_t))$. $\psi_t$ is a feature extracted from a bi-directional LSTM $\psi(R)$. Since the module indicator keeps unchanged within a segment, the posterior distribution $q_{\phi_z}(z_t|M, R)$ is conditioned on the boundary indicator $m_{t-1}$ and defined as:

$$q_{\phi_z}(z_t|M, R) = m_{t-1} \cdot \tilde{q}(z_t|\psi_t) + (1 - m_{t-1}) \cdot \delta(z_t = z_{t-1}), \tag{8}$$

where the transition function $\tilde{q}(z_t|\psi_t)$ is implemented as a multinomial distribution parameterized by $\mathrm{Softmax}(f'_{z-\mathrm{mlp}}(\psi_t))$. Once we have the posterior distribution, we apply Gumbel-Softmax[10] with straight-through estimators[2] to take samples of $m_t$ and $z_t$.

**Weak Supervision on M and Z.** We first use StanfordNLP toolkit [17] to parse every response in the training set as a sequence of segments, and use $\tilde{M} = \{\tilde{m}_t\}_{t=1}^{l_r}$ to denote the results of segmentation labeling. The pseudo label of module choice $\tilde{Z} = \{\tilde{z}_t\}_{t=1}^{l_r}$ is tagged in a similar way to multiclass classification, determined by (1) the similarity between each segment and knowledge and (2) the classification confidence of the style discriminator. More details about the construction of $\tilde{Z}$ and $\tilde{M}$ are provided in the supplementary material.

With $\tilde{Z}$ and $\tilde{M}$, the loss function of weak supervision is defined as:

$$\mathcal{L}_m = -\sum_{t=1}^{l_r} \log p_{\theta_m}(\tilde{m}_t|r_{\leq t}, \tilde{z}_t),$$
$$\mathcal{L}_z = -\sum_{t=1}^{l_r} \tilde{m}_{t-1} \cdot \log p_{\theta_z}(\tilde{z}_t|r_{<t}, \tilde{z}_{<t}, \tilde{m}_{t-1}). \tag{9}$$

The learning algorithm is summarized in the supplementary material.

## 4 Experiments

### 4.1 Datasets

We test our model on benchmarks of knowledge-grounded dialogue generation, including Wizard of Wikipedia (Wizard) and CMU Document Grounded Conversations (CMU_DoG) [49]. Both datasets are split into training sets, validation sets, and test sets by the data owners. Topics in Wizard cover a wide range ($1,365$ in total), and each conversation happens between a wizard who has access to the knowledge about a specific topic and an apprentice who is just eager to learn from the wizard about the topic. The test set is split into two subsets: Test Seen and Test Unseen. Test Seen only contains dialogues with topics that have already appeared in the training set, while topics in Test Unseen never appear in the training set and the validation set. We follow [3] and conduct the pre-processing with the code published on ParlAI[2]. Different from Wizard, CMU_DoG focuses on movie domain, and

---

[1]We always have $m_0 = 1$

[2]https://github.com/facebookresearch/ParlAI/blob/master/projects/wizard_of_wikipedia

besides wizard-apprentice conversations, the data also contain conversations between two workers who know the document and try to discuss the content in depth. In both datasets, only the turns where knowledge is accessible are considered in response generation. More details are described in the supplementary material.

We choose the Reddit Corpus published by [15] as $\mathcal{D}$. The data contains $842,521$ context-knowledge-response triples for training and $2,737$ context-knowledge-response triples for validation. On average, each dialogue contains 3.1 utterances in both sets, and the average length of the utterance is 16.0 in training and is 16.1 in validation. The dataset enjoys a great diversity of expression styles thanks to the large scale of corpus and little restriction on expression. We use part of the training data of Wizard and CMU_DoG as $\mathcal{D}_{sty}$ respectively, for these two datasets are distinctive in expression style and differ from each other. The dialogues in CMU_DoG tend to be causal and short, with most utterances irrelevant to knowledge. While the responses in Wizard are usually long and knowledgeable, as some phrases are directly extracted from wiki articles.

## 4.2 Experimental Setup

In this paper, we mainly consider two experimental setups, corresponding to the two subproblems mentioned in Sec 3.1. To explore how our model can be used to control the distribution of different kinds of segments (knowledge-related and knowledge-irrelevant), we first train the model on the Reddit Corpus and then fine-tune it on a small amount of examples in Wizard and CMU_DoG, respectively. To verify whether our model can generate the knowledge-related segments in the desired style, we still train the model on the Reddit Corpus, and use a style tag to control the generation process. In this experimental setup, we are primarily concerned with generating with two kinds of styles, positive and negative, where $z_t = 2 \cdot \min(1, z_t)$ tells the model to generate a response in positive sentiment and $z_t = 3 \cdot \min(1, z_t)$ is for response in negative sentiment.

**Evaluation Metrics.** We choose distinct and unigram F1 [3] as metrics, where the F1 metric is calculated with the code published at `https://github.com/facebookresearch/ParlAI/blob/master/parlai/core/metrics.py`. Distinct-1 (D-1) and Distinct-2 (D-2) are calculated as ratios of distinct unigrams and bigrams in responses, respectively. We also employ classification accuracy as the evaluation metrics for style control experiments. Specifically, we exploit Roberta trained on the SST-2 training set [27] as the evaluator, which is more accurate than that from the classifiers in [46].

**Baselines.** For the exploration of the first subproblem, we select the following models as baselines: (1) **BART**[12]: a model that achieves state-of-the-art performance on various text generation tasks. Note that our model degrades into BART once we remove the module indicator Z and the boundary indicator M; (2) **Zero-resource Knowledge-grounded Conversation (ZRKGC)** [15]: [3] a model that is based on UniLM [4] and optimized with Generalized EM method. The model is trained on the Reddit Corpus and achieves comparable performance with state-of-the-art methods that rely on knowledge-grounded dialogues for training. For the second subproblem, we consider the following models as baselines: (1)**Emotional Chatting Machine (ECM)**[47]: [4] a model which can generate appropriate responses not only content-relevant but also emotional consistent; (2)variant of **DialoGPT**[42]: DialoGPT is a model that is pre-trained on large-scale conversation corpus and attains a performance close to human in single-turn dialogues. As DialoGPT is not designed for sentiment control, we add a sentiment indicating token at the first of the sequence and explore whether such simple heuristics works for controlling knowledge expression. Comparisons with more state-of-the-art models are provided in the supplementary material.

## 4.3 Results on Learning Structure Style

In this section, we demonstrate the effectiveness of our segmentation-based generation framework in both low-resource setting and zero-resource setting and empirically verify that our model can learn structure style with a few annotated examples. In zero-resource setting, we trained our model on the Reddit Corpus published by [15] and tested on Wizard and CMU_DoG respectively. Automatic evaluation results are shown in Table 2. It could be observed that: (1) our model significantly

---

[3] `https://github.com/nlpxucan/ZRKGC`
[4] `https://github.com/thu-coai/ecm`

Table 2: Automatic evaluation results. Numbers in bold mean that the improvement to the best performing baseline is statistically significant (t-test with $p$-value < 0.05).

| Training Data | Models | Wizard Seen | | | Wizard Unseen | | | CMU_DoG | | |
|---|---|---|---|---|---|---|---|---|---|---|
| | | F1 | D-1 | D-2 | F1 | D-1 | D-2 | F1 | D-1 | D-2 |
| Reddit Corpus | BART | 18.4 | 0.076 | 0.355 | 18.4 | 0.049 | 0.237 | 9.8 | 0.021 | 0.131 |
| | ZRKGC | 18.9 | 0.055 | 0.246 | 18.8 | 0.037 | 0.179 | 12.2 | 0.015 | 0.094 |
| | Our Model | **19.3** | **0.082** | **0.383** | **19.2** | **0.060** | **0.292** | 12.2 | **0.028** | **0.186** |
| Reddit Corpus + 10% annotated data | BART | 18.9 | 0.073 | 0.357 | 18.8 | 0.049 | 0.235 | 10.1 | 0.019 | 0.110 |
| | ZRKGC | 19.1 | 0.072 | 0.309 | 18.9 | 0.048 | 0.209 | 13.7 | 0.010 | 0.062 |
| | Our Model | **20.4** | 0.073 | **0.366** | **20.0** | 0.052 | **0.270** | **14.4** | 0.015 | **0.122** |

(a) Wizard Seen  (b) Wizard Unseen  (c) CMU_DoG

Figure 2: Performance of different models wrt. training data size.

outperforms ZRKGC and BART on most metrics and achieves the new state-of-the-art performance on Wizard. It is impressive that our model exceeds BART in CMU_DoG especially since the proposed model degrades into BART without two sequential latent variables Z and M. The result serves as strong evidence for the effect of two latent variables, which enable the model to learn complex expression style in Reddit Corpus to handle flexible expression in CMU_DoG. By contrast, BART is far from satisfying with only a regular decoder. (2) our model exceeds ZRKGC significantly in terms of Distinct metrics, for ZRKGC mainly focuses on leverage external knowledge sources for response generation, but falls short on expression diversity. In low-resource setting, after training our model on the Reddit Corpus, we then fine-tune it with only 10% training size of Wizard and CMU_DoG respectively (i.e., $\mathcal{D}_{sty}$ in Sec 3.1) to adjust $p(z_t)$ and $p(m_t)$ to a new structure style. When provided with only 10% training data, our model gets obvious improvement (~ 1% increase in F1) in contrast with BART (~ 0.5% increase in F1) and ZRKGC (~ 0.2% increase in F1), proving that the proposed model can learn more sophisticated structure style through quickly adjustment on a specific dataset with little cost. Furthermore, we are interested in its potential in learning with less annotated data. We also want to investigate how our model is adjusted to different annotated data. Exploration of these two topics is as follows.

**Fine-tune with less annotated data.** We first train the model on the Reddit Corpus and then fine-tune it with the amount of annotated data(e.g., Wizard and CMU_DoG) gradually increasing from 2% to 10%. To have a more intuitive understanding of the effects of latent variables Z and M, we compare the proposed model with BART, which generates the response with a single decoder. The evaluation results are shown in Figure 2. It can be concluded from the result that: (1) our model can learn the expression style of a particular dataset more efficiently. As the training data increase, our model has a more significant improvement in terms of the F1 metric; (2) our model performs better in meager resources since there is a considerable gap between our model and BART when the training data is close to 0%; (3) the expression style of CMU_DoG can be learned with less data because the model has a significant change in performance after using 2% CMU_DoG training data.

**Refashioning of knowledge-related segments.** To know how our model adjusts to different datasets, we compare the knowledge-related segments before and after trained with annotated data from two aspects: (1) the average proportion of knowledge-related segments ($pklg$) in a sentence; (2) the average proportion of words belonging to knowledge-related segments ($lklg$). Figure 3 reports the results. The results indicate that our model could learn the underlying structure style of both datasets, with the great difference of $pklg$ and $lklg$ before and after fine-tuning as evidence. After fine-tuned with Wizard data,

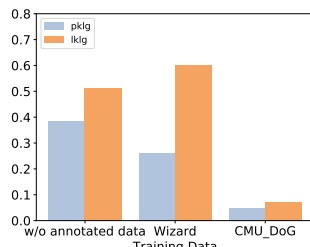

Figure 3: The effect of fine-tuning on different data.

$pklg$ drops to $0.26$ while the $lklg$ grows up a bit, indicating that the knowledge-related segments generated by our model are fewer and longer, which tallies with the fact that the responses in Wizard are probably directly copied from background knowledge. However, after CMU_DoG data is fed to the model, both $pklg$ and $lklg$ shrinks drastically, which agrees with the fact that crowd-sourcing workers converse more liberally online and the responses are less relevant to background knowledge.

## 4.4 Results on Learning Content Style

Table 3: Evaluation results on sentiment control. Numbers in bold mean that the improvement to the best performing baseline is statistically significant (t-test with $p$-value $< 0.05$).

| Models | Wizard Seen | | | | Wizard Unseen | | | | CMU_DoG | | | |
| | positive | | negative | | positive | | negative | | positive | | negative | |
| | F1 | Acc | F1 | Acc | F1 | Acc | F1 | Acc | F1 | Acc | F1 | Acc |
|---|---|---|---|---|---|---|---|---|---|---|---|---|
| ECM | 10.5 | 55.8 | 10.2 | 60.7 | 10.1 | 55.7 | 10.1 | 57.6 | 7.6 | 41.5 | 8.3 | 55.4 |
| DialoGPT | 12.1 | 54.1 | 12.1 | 46.9 | 12.0 | 56.0 | 12.0 | 45.0 | 9.2 | 44.9 | 9.2 | 55.1 |
| Our Model | **19.7** | **70.3** | **19.2** | **70.7** | **19.4** | **73.1** | **19.2** | **69.9** | **12.7** | **74.8** | **12.2** | **68.0** |

We further investigate whether the proposed model could express knowledge with the desired sentiment. Specifically, we introduce two sets of style adapters to endow knowledge expression in two different sentiments, namely positive and negative. So in this scenario, it is required that responses are not only coherent with context but also limited in positive or negative sentiment. To apply ECM on knowledge-grounded conversation, we label the sentiment category for each response with a classifier pre-trained on the SST [27] training set. For DialoGPT, we similarly annotate each response with a sentiment category and append the sentiment token before the context tokens. The evaluation results is shown in Table 3. We can conclude that: (1) The proposed model outperforms all baseline models in terms of all metrics, which indicates that our model can control the sentiment of knowledge expression and guarantee high quality of the generated responses; (2) Simply adding a sentiment indicating token at the beginning of the sequence can not effectively control the style of knowledge expression, as the performance of DialoGPT on sentiment control is poor; (3) Although ECM is designed for sentiment control, it still fails to perform well in this task, proving that sentiment control in the knowledge-grounded conversation is rather difficult. Besides, ECM can only control the sentiment of the whole response but is helpless to manage every knowledge-related segment at a more refined level.

## 5 Conclusions

We explore knowledge expression in knowledge-grounded conversation and break down the expression style of a response into the structure of the response (structure style) and the style of the content in each part (content style). We propose a variational segmentation-based generation model to discover the underlying expression style in response. Specifically, we introduce two latent variables to model these two aspects of expression style respectively and induce an evidence lower bound of the likelihood. Evaluation results on two benchmarks of the task indicate that our model can learn the structure style with little cost and generate responses in desired content style without any human-annotated data.

## Broader Impact

Enabling an open-domain dialogue system to automatically detect and discover the underlying structural pattern of a sentence is of great significance. This process is destined to be hailed as a milestone on the way to thoroughly reveal the essential nature of open-domain dialogue. Capable of handling different expression styles, positive or negative, casual or serious, our work implies that we are now much closer to the final destination of constructing an artificial intelligent dialogue system that could communicate freely with human being, which is beyond the wildest dream of most AI and NLP researchers. In the future, we heartily look forward to seeing advanced methods or ideas based on our work, and we expect the appearance of related industrial projects and applications to benefit the people and the public.

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
