# OpenReview forum: "Learning to Express in Knowledge-Grounded Conversation"
_NeurIPS.cc/2021/Conference — NeurIPS 2021 Submitted_

### Official Review · Reviewer_gpDA · 2021-07-14

**Rating:** 5
**Confidence:** 3

**Summary:**

This paper explores the knowledge expression in knowledge-grounded conversation on two benchmark datasets - Wizard of Wikipedia(Wizard) and CMU Document Grounded Conversation(CMU_DoG). The authors propose a novel variational segmentation-based model to control and capture the expression style by using a few examples. The proposed model outperforms the SOTA on both datasets.


**Limitations And Societal Impact:**

Yes / NA

**Main Review:**

Strengths:
- The introduction of the latent variables to capture the style is reasonably intuitive and well-motivated.
- Relevant baselines have been used for this task.
- SOTA on both the datasets, which were also found to be statistically significant for few-shot learning.
- The authors provide the corresponding code and detailed description of hyperparameters which would be essential for reproducibility if open-sourced.

Weakness:

- The paper is not well-written and lacks details making it difficult to understand; see comments below.
- It is not clear if the model uses pre-trained BART or just the model architecture as the backbone (line 141, 262). If the latter, it is not clear why the pre-trained weights were not considered.
- If a pre-trained BART is used, it seems it would create a mismatch between pre-training and fine-tuning, since the [EOS] token does not appear during pre-training. Could the authors confirm if the [EOS] token is used as a separator and not the [SEP] token in equation 1.
- Pre-training on Reddit corpus is also not adequately motivated. Why was this particular dataset used for pre-training and the similarity of this dataset with the downstream tasks? An ablation study depicting the model trained from scratch on the respective datasets could provide more information if the improvement is because of more training data or the introduction of latent variables.
- An ablation study is required to understand the importance of each of the latent variables introduced for this task.
- Detailed information about the adapters in the decoder is missing. Figure 1 could also be improved by showing the decoder as well as the adapters used. An ablation study with or without the adapters in the decoder layer is necessary to understand their importance. Also, it should be explicitly mentioned if all the models in Tables 2 and 3 use these adapters.
- While the authors claim that the model generates the responses, it is unclear if it is indeed an autoregressive model and the output is generated word by word. The corresponding loss function could provide more details. If the whole task is a generation setup, corresponding NLG metrics are not considered. Some model-generated examples would provide more insight.
- Y-axis for Figure 2 is misleading and provides a hyperbole of the findings. It is not clear if the authors also experimented by incorporating 100% training data and how the results changed.
- Pklg and lklg are not properly motivated as metrics and their correct interpretation is not provided. Relevance of background knowledge would be well justified by corresponding data analysis.


Questions:
- On line 146, how do the authors handle sequences of variable length? Are all the sequences of $l_{k}$? Do the authors use padding?
- What does delta in Equation 2 and 8 signify? Does it refer to the Dirac delta function?
- Similarly, could the authors correctly specify what does f mean on line 205 and 210

Suggestions/Comments:
- Please be careful of the formatting. Eg. line 134, 136, 268, 298
- Line 40: Structure style -> The structure style
- Line 41: even the -> even if the
- Line 72: Exploration the -> Exploration of the
- Line 113: each is other -> each is either
- Line 204: Base -> Based
- Line 206: keeps -> is kept
- Line 287: leverage -> leveraging
- Line 288: In low -> In the low
- Line 293: quickly -> quick
- Line 316: After fine-tuned -> fine-tuning
- Line 321: to -> to the
- Please provide the y-axis for Figure 3.

The results seem promising, however, the paper could be improved in the presentation and more detailed descriptions.

Update after rebuttal

Thank you authors for answering my questions. Though the results seem promising, I believe the paper needs restructuring a bit and more clarification about the experimental setup as well as the human evaluations. Maintaining the original rating.

**Time Spent Reviewing:**

8-10

---

> ### Author Response · Authors · 2021-08-10
> **Respond to Reviewer gpDA**
>
> Q1: About the writing.
>
> A1: Thanks for your valuable comments, and we will revise the paper according to your advice.
>
>
>
> Q2: It is not clear if the model uses pre-trained BART or just the model architecture as the backbone.
>
> A2: We use the pre-trained BART and will clarify this point in the final draft.
>
>
>
> Q3: Could the author confirm if the [EOS] token is used as a separator and not the [SEP] token in equation 1.
>
> A3: Designers of the BART do not provide relevant implement details in the paper about which symbol is used when pre-training the model. We exploit the pre-trained model provided by Hugging Face. It uses <s> and </s> to denote the beginning and the end (or separation) of a sequence when pre-training. And we also exploit the corresponding tokenizer by Hugging Face to preprocess the input to the model, which adds <s> and </s> by default when tokenizing a long sentence. So the mismatch between the pre-trained model and the fine-tuning does not exist actually.
>
>
>
> Q4: Why was this particular dataset used for pretraining and the similarity of this dataset with the downstream tasks? An ablation study depicting the model trained from scratch on the respective datasets could provide more information if the improvement is because of more training data or the introduction of latent variables.
>
> A4: Because the Reddit corpus is constructed from an online forum and abundant in expression styles, the two variables could be properly initialized. In addition, the two baselines(i.e., BART and ZRKGC) are also trained on the same dataset, which gives a fair comparison and demonstrates the effectiveness of introducing latent variables.
>
>
> Q5: An ablation study is required to understand the importance of each of the latent variables introduced for this task.
>
> A5: Please refer to Section 7 in the supplementary material for more details about Ablation over Weak Supervision.
>
>
>
> Q6: Detailed information about the adapters in the decoder is missing. Figure 1 could also be improved by showing the decoder as well as the adapters used. An ablation study with or without the adapters in the decoder layer is necessary to understand their importance. Also, it should be explicitly mentioned if all the models in Tables 2 and 3 use these adapters.
>
> A6: Thanks for your valuable advice, the implementation details of adapters are identical to [1]. We will claim it in the final version.
> We are sorry but no such 'ablation study' about the adapters could be conducted. They are an indispensable part of our model and we rely on adapters to generate segments in different structure styles. They are the executors of the decisions made by the module indicator and the boundary indicator. If removed, a standard decoder has no way to utilize the segmentation inference by the latent variables.
>
> [1] Parameter-efficient transfer learning for nlp
>
>
>
> Q7: It is unclear if it is indeed an autoregressive model and the output is generated word by word.
>
> A7: The model is still auto-regressive since the generation at each step still depends on the previously generated words(in line 182). Words within a segment are not generated simultaneously but dependent on the word generated before.
>
>
>
> Q8: If the whole task is a generation setup, corresponding NLG metrics are not considered.
>
> A8: The F1 metric is one of the most important metrics for knowledge-grounded dialog generation and is widely adopted by many works. However, we also added other nlg metrics(BLEU, METEOR, and ROUGE) as follows:
>
>
> | Training Data           | Models    | Wizard Seen |        |         | Wizard Unseen |        |         | CMU_DoG |        |         |
> |-------------------------|-----------|-------------|--------|---------|---------------|--------|---------|---------|--------|---------|
> |                         |           | BLEU-1      | METEOR | ROUGE-L | BLEU-1        | METEOR | ROUGE-L | BLEU-1  | METEOR | ROUGE-L |
> | Reddit Corpus           | BART      | 0.206       | 0.099  | 0.164   | 0.202         | 0.099  | 0.167   | 0.141   | 0.060  | 0.117   |
> |                         | ZRKGC     | 0.225       | 0.099  | 0.163   | 0.220         | 0.100  | 0.164   | 0.161   | 0.075  | 0.128   |
> |                         | Our Model | 0.235       | 0.095  | 0.168   | 0.232         | 0.095  | 0.169   | 0.166   | 0.069  | 0.131   |
> | Reddit Corpus+10\% data | BART      | 0.203       | 0.103  | 0.174   | 0.199         | 0.103  | 0.175   | 0.141   | 0.067  | 0.122   |
> |                         | ZRKGC     | 0.227       | 0.101  | 0.175   | 0.222         | 0.100  | 0.176   | 0.173   | 0.083  | 0.139   |
> |                         | Our Model | 0.237       | 0.103  | 0.181   | 0.231         | 0.101  | 0.176   | 0.182   | 0.080  | 0.139   |
>
> Q9: Some model-generated examples would provide more insight
>
> A9: Please refer to Section 9 'Case Study' in the supplementary material.
>
>
>
> Q10: Y-axis for Figure 2 is misleading and provides a hyperbole of the findings. It is not clear if the authors also experimented by incorporating 100\% training data and how the results changed.
>
> A10: Thanks for your valuable advice. We use a small amount of data to verify the practicality of our model in real scenarios where the annotated data is challenging to obtain. Additionally, we provide the F1 score on the whole training data as follows:
>
> |               | Models    | Wizard Seen | Wizard Unseen | CMU_DoG |
> |---------------|-----------|-------------|---------------|---------|
> | (w/o Reddit)  | Our Model | 22.0        | 20.5          | 15.3    |
> | (With Reddit) | Our Model | 21.9        | 21.2          | 15.5    |
>
>
>
> Q11: Pklg and Lklg are not properly motivated as metrics and their correct interpretation is not provided. Relevance of background knowledge would be well justified by corresponding data analysis.
>
> A11:Maybe we haven't made it clear enough in the paper. Please refer to lines 310-313 for the interpretation of Pklg and Lklg, and refer to Section 8 Human Evaluation in the supplementary material for the evaluation of knowledge relevance.
>
>
>
> Q12: how do the authors handle sequences of variable length?
>
> A12: The input sequence is padded to the maximum input length, which is 210 in our model. Sequence longer than that is truncated. Annotation about the process of variable input length will be added in the final draft.
>
>
>
> Q13: What does delta in Equation 2 and 8 signify? Does it refer to the Dirac delta function?
>
> A13: Yes, It denotes the Dirac delta function. The probability is all zero except an impulse where $z_t$ is equal to $z_{t-1}$. We will make it more clear in the final draft.
>
>
>
> Q14: could the authors correctly specify what does f mean on line 205 and 210?
>
> A14: It denotes a multi-layer perceptron network as indicated by its subscript.

---

> ### Author Response · Authors · 2021-08-28
> **About the human evaluation**
>
> Thank you for taking the time to read our responses and giving the new comments and suggestions.  Actually, we have conducted a **THOROUGH** human evaluation and presented the results in the Supplementary Material. Could you please clearly point out what's the problem with the human evaluation and the experiment setup? We could reconstruct a bit more and be more detailed.

---

### Official Review · Reviewer_Hbvb · 2021-07-15

**Rating:** 7
**Confidence:** 4

**Summary:**

The authors study the problem of knowledge-grounded conversation, and propose a segment generation process to better “express” knowledge in the conversation. Specifically, the authors identify that expressing a response can be broken down into two components: the structure of the response and the content style; the former marked by either “knowledge-related” or “knowledge-irrelevant” segments, while the latter is characterized by positive, negative, or other styles of formulating a response. The generation model is based on a normal encoder/decoder architecture, but the decoder is influenced by various introduced latent variables when determining how to generate the next token. A “boundary” indicator informs the decoder when it has reached the boundary of a segment (and must therefore transition), and a “module” indicator determines which structure/segment type to use when generating; these indicators inform the decoder which encoded representations to attend to when generating and are learned via weak supervision. The authors conduct experiments on two knowledge-grounded datasets and show that their model outperforms strong baselines in both zero-resource and low-resource settings, and additionally show that their method is more adaptable to training data (as it can learn more in a low-resource setting than the other models). The authors additionally show that the model is more controllable in a sentiment-control setting, such that sentiment classifiers indicate the controlled responses more effectively model a positive or negative sentiment depending on the setup.

**Limitations And Societal Impact:**

As noted in the main review, I would have liked to have seen some justification for not applying the style adapters in the knowledge-irrelevant regime (even if it’s as simple as maintaining elegance/less computational costs)

**Main Review:**

*Originality*: The method is original and is an interesting approach to response generation (specifically the segment generation component). Related work is adequately cited and it is clear how the work differs from prior contributions; it is important to note that the authors study explicitly the **expression of knowledge**, rather than the selection of knowledge, and thus compare to methods trained in similar fashions (specifically the ZRKGC method for WoW and CMU_DoG)

*Quality*: The submission is technically sound, and claims are well supported via experimental results; the authors display a few interesting analyses of how the model learns (fine-tuning on less annotated data and refashioning of knowledge-related segments). However, without an analysis of the qualitative model outputs, I am a little bit skeptical about the segment generation process → how are we sure that the models responses are coherent if the model is switching between various modules? F1 word overlap only measures so much.

*Clarity*:
- The paper is well-written and methods/results are clearly outlined and presented in the text.
- However, the “content style” component of the method is less studied/explained than I would have liked. To start, the majority of the analysis focused explicitly on knowledge expression, but this left out more analysis on the sentiment control, and so that section felt a bit lacking. Additionally, it is not particularly clear what “adapters” were inserted into the decoder layers when in the stylized knowledge regime. Finally, it is not clear why the authors chose to limit their study of stylized generation to the knowledge-relevant sections only; could this have brought additional control over sentiment if there was stylized generation for the knowledge-irrelevant sections?

*Significance*: The authors obtain SOTA results on the two datasets compared to similarly trained baselines in low-resource settings, and indeed the methods appear to learn quicker from less data, although it would have been interesting to see the model’s performance when trained on the full set of data.


***EDIT Following Discussion Period***

After reading other reviews (and further discussion with other reviewers), I am revising my score to be 6. I agree with others' sentiments that a more thorough evaluation is required to adequately measure the efficacy of the method (e.g., a human evaluation), and indeed in its current state it is unclear what the true motivation/focus of the paper is (few/zero-shot knowledge grounding or knowledge grounding in general).


***2nd EDIT Following Discussion Period***

The authors have done an appropriate job of addressing other reviewers concerns (as well as mine) in further discussion, so I am inclined to revise my score back to a 7. I thank the authors for their willingness to engage in discussion and provide additional details where necessary.

**Time Spent Reviewing:**

2.5 hours

---

> ### Author Response · Authors · 2021-08-10
> **Respond to Reviewer Hbvb**
>
> Q1: Without an analysis of the qualitative model outputs, and how are we sure that the models responses are coherent if the model is switching between various modules?
>
> A1: The generation of each module is conditioned on all previously generated text to ensure fluency, as indicated in Section 3.2. In addition, we also conduct human evaluation and provide an example in the supplementary, which also indicates that the proposed model can guarantee smooth switching between various modules.
>
>
>
> Q2: the majority of the analysis focused explicitly on knowledge expression, but this left out more analysis on the sentiment control, and so that section felt a bit lacking.
>
> A2: We additionally add two baselines(i.e., CTRL and BART) for sentiment control, the results are as follows:
>
> | Models    | Wizard Seen |      |          |      | Wizard Unseen |      |          |      | CMU_DoG  |      |          |      |
> |-----------|-------------|------|----------|------|---------------|------|----------|------|----------|------|----------|------|
> |           | positive    |      | negative |      | positive      |      | negative |      | positive |      | negative |      |
> |           | F1          | Acc  | F1       | Acc  | F1            | Acc  | F1       | Acc  | F1       | Acc  | F1       | Acc  |
> | BART      | 15.4        | 68.3 | 15.1     | 50.2 | 15.3          | 69.3 | 14.9     | 49.2 | 9.8      | 66.9 | 9.7      | 55.4 |
> | CTRL      | 15.3        | 71.9 | 14.9     | 55.3 | 14.9          | 75.0 | 14.6     | 52.3 | 9.3      | 70.2 | 9.4      | 61.7 |
> | Our Model | 19.7        | 70.3 | 19.2     | 70.7 | 19.4          | 73.1 | 19.2     | 69.9 | 12.7     | 74.8 | 12.2     | 68.0 |
>
> The experimental results indicate that our model still outperforms CTRL and BART on sentiment control. In addition, CTRL and BART tend to generate sentences with a positive sentiment mainly due to the imbalanced training data distribution.
>
>
>
> Q3: It is not particular clear what "adapters" were inserted into the decoder layers when in the stylized knowledge regime.
>
> A3: As illustrated in lines 13-14 of Algorithm1 in the supplementary material, we pre-train the adapters for the styled knowledge regime(i.e., positive and negative) based on the likelihood term in ELBO. And the adapters used in our model are implemented identically to [1].
>
> [1] Parameter-efficient transfer learning for nlp
>
>
>
> Q4: It is not clear why the authors chose to limit their study of stylized generation to the knowledge-relevant sections only?
>
> A4: Due to the segmentation, we can easily add sentiment control on the knowledge-irrelevant section, but this does not contribute much to the knowledge expression, which is the main problem we want to resolve in this work. Thanks for your suggestions and we are ready to explore the expression of knowledge-irrelevant sections in our future work.
>
>
>
> Q5: Some justification for not applying the style adapters in the knowledge-irrelevant regime
>
> A5: Please refer to Answer for Q4.

---

> ### Author Response · Authors · 2021-08-26
> **Response to "EDIT Following Discussion Period"**
>
> We sincerely appreciate your positive comments.
>
>
> - ***Human evaluation***
>
> We have conducted a human evaluation to measure the efficacy of the proposed model from the following four aspects: ***Fluency***, ***Context Coherence***, ***Knowledge Relevance***, and ***Style Consistency***. The first three aspects are widely adopted by previous works about knowledge-grounded dialog[1][2] and the final aspect is used to judge the quality of style expression. Due to space limitations, we provide these results in the Supplementary Material (Table 4 and Table 5).  From the evaluation results, we can observe an acceptable Kappa  and find that our model outperforms the baselines methods in terms of four aspects.
>
> The human evaluation suggested by reviewer MMWY is only to measure whether the model conforms to ethics rather than its efficacy. We acknowledge that faithfulness to knowledge is a common issue for most of the current end-to-end knowledge-grounded dialog systems.  In contrast, our proposed model enjoys better controllability and can be extended to aid this problem with the following modifications:  (1) we could strictly control the source of knowledge sentences. (2) we could define ''entailed'' as a new content style to generate a response that is semantically entailed by the source knowledge. More details can be found in A4 in the second response to reviewer MMWY.
>
>
>
> - ***Clarification of our motivation and focus***
>
> In this paper, we mainly focus on the stylized knowledge expression in the knowledge-grounded dialog. However, it isn't easy to obtain the stylized KGC data in real scenarios, and this motivates us to conduct few-shot experiments, which are more practical and challenging. Even so, we also present the results about using the whole dataset (in A5 in response to reviewer MMWY) and prove that our proposed model outperforms KnowledGPT (a state-of-the-art model in knowledge-grounded dialog[3]). We will revise the introduction to clarify this point.
>
> Thanks for your valuable comments again. We hope our clarification can address your concerns, and we will revise our paper to make it more clear.
>
>
>
> [1] Zero-Resource Knowledge-Grounded Dialogue Generation
>
> [2] Low-Resource Knowledge-Grounded Dialogue Generation
>
> [3] Knowledge-Grounded Dialogue Generation with Pre-trained Language Models

---

> ### Author Response · Authors · 2021-08-28
> **About the human evaluation**
>
> Thanks again for taking the time to read our responses and giving the new comments and suggestions. Actually, we have conducted a **THOROUGH** human evaluation and presented the results in the Supplementary Material. Could you please clearly point out what's the problem with the human evaluation? We could reconstruct a bit more and be more detailed.

---

> ### Author Response · Authors · 2021-09-01
> **Clear up the possible misunderstanding & seek support**
>
> Thanks for your constructive comments and valuable suggestions in the rebuttal stage.  We are very sorry that you turned down the rating of our work possibly due to some concerns about human evaluation. To clear up your possible misunderstandings, we would like to clarify once again that a detailed human evaluation has been already conducted and the results have been presented in our supplementary material (Table 4 and Table 5).
>
> Besides, hdNj and MMWY, the two reviewers who put up the concerns of human evaluation, have acknowledged our clarifications and made new decisions about their ratings. We are very appreciative and encouraged by that.
>
> We sincerely hope that you can check our human evaluation in the appendix at your convenience. We will definitely move this part into the main document as a subsection and will also improve our introduction to clarify our motivation/focus in our final version.
>
> Especially, we need your support during the discussion phase. We will be extremely grateful if you could maintain your original rating. It means a lot to us.

---

> > ### Comment · Reviewer_Hbvb · 2021-09-10
> > **Response**
> >
> > I appreciate your reaching back out to continue this discussion. After reading your detailed responses and the other reviewers' responses to those, I am inclined to revise my score back up to a 7; I have updated my review to reflect this.

---

> > > ### Author Response · Authors · 2021-09-10
> > > **Thank you**
> > >
> > > We are grateful for your positive feedback. It's encouraging! We will incorporate your suggestions in the final version.

---

### Official Review · Reviewer_MMWY · 2021-07-16

**Rating:** 6
**Confidence:** 5

**Summary:**

This paper tackles the knowledge grounded dialogue generation task. The paper primarily explores two aspects of knowledge expression: (1) structure of the response, and (2) style of the content. Hence, the proposed model introduces two sequential latent variables to pre-trained encoder-decoder models like BART. The paper proposes a segmentation based generation model which first predicts if the current token to be generated belongs to the previous segment or not and if not then it predicts the type of the new segment. Segment types are knowledge-related (conditions on the knowledge) or knowledge-irrelevant (conditions on the dialogue context).

The paper proposes a novel model that explores the expression of knowledge in grounded conversations. The proposed model leverages pre-trained encoder-decoder models and extends their ability by adding two latent variables: one for representing structure and the other for style. Additionally, the proposed model has a variational segmentation-based decoder.


**Limitations And Societal Impact:**

The “Broader Impact” section does not talk about the negative impact of this technology. Overall, knowledge grounded conversation models have the ability to spread fake news. Especially the proposed model which can perform well in zero-shot settings. If the document that is used for grounding is not factually correct, then the model will generate factually incorrect/fake responses. Additionally, the proposed model has the capability of controlling style (positive/negative is reported in the paper). This can be used to further tune the responses according to the target audience.

**Main Review:**

**Originality:** Knowledge grounded conversation response generation is a well-established task. The paper proposes a novel idea of using knowledge-related or knowledge-irrelevant segments to generate responses. The proposed model is novel especially because it does not rely on using the vanilla pre-trained encoder-decoder model but suggests a useful extension to the model by introducing two latent variables and optimizing the model using a variational approach. Additional papers to cite are:

[1*] ​​Ghazvininejad, Marjan, Chris Brockett, Ming-Wei Chang, Bill Dolan, Jianfeng Gao, Wen-tau Yih, and Michel Galley. "A knowledge-grounded neural conversation model." In Proceedings of the AAAI Conference on Artificial Intelligence, vol. 32, no. 1. 2018.

[2*] Prabhumoye, Shrimai, Kazuma Hashimoto, Yingbo Zhou, Alan W. Black, and Ruslan Salakhutdinov. "Focused Attention Improves Document-Grounded Generation." In Proceedings of the 2021 Conference of the North American Chapter of the Association for Computational Linguistics: Human Language Technologies, pp. 4274-4287. 2021.

[3*] Shuster, Kurt, Spencer Poff, Moya Chen, Douwe Kiela, and Jason Weston. "Retrieval Augmentation Reduces Hallucination in Conversation." arXiv preprint arXiv:2104.07567 (2021).

**Quality:** Although the proposed model is exciting, the paper lacks a thorough evaluation. The paper has reported results only on F1 and D1/D2 metrics. Human evaluation is reported in the supplementary material but it is performed only by 3 annotators. The paper does not report Bleu, METEOR or Rouge metrics which are very important to evaluate generated text. The paper had the opportunity to perform further interesting fine-grained evaluation such as accuracy of predicting segment boundaries and accuracy of predicting segment types to better understand the contribution of the two latent variables. The paper also does not report numbers by training the proposed model on the whole Wizard/CMU_DoG dataset. Since the focus of the paper is to solve knowledge-grounded conversations, this result would add value to the paper. If the focus of the paper is only on few-shot/zero-shot cases then this should be made more explicit in title, abstract and introduction. Overall, it is not clear if the proposed model is faithful to the knowledge provided in the documents/paragraphs. Given the lack of a comprehensive evaluation, it is hard to appreciate the performance of the model in comparison with others.

**Clarity:** The paper is clearly written. More details about implementation, algorithm as well as a case study is provided in the supplementary section.

**Significance:** In Table 2, we see a higher gain for the ZRKGC model for D1/D2 metrics on the Wizard dataset when going from zero-shot case to 10%-annotated case. Hence, the claim on Line 293 may need to be revisited.

**Review after Rebuttal**

The authors have promised to add details about additional automated metrics, revised ethics statement, results of additional human evaluation which measures the faithfulness of the generated response to the knowledge and a clarification of their goal in the introduction. I am leaning towards acceptance if these changes are made. I have updated my score to reflect the same.

**Time Spent Reviewing:**

5

---

> ### Author Response · Authors · 2021-08-10
> **Respond to Reviewer MMWY**
>
> Q1: Additional papers to cite.
>
> A1: Thanks for your valuable advice. We will revise the final draft.
>
>
>
> Q2: Human evaluation is performed only by 3 annotators.
>
> A2: We follow [1] and [2] to perform Human Evaluation and recruit three annotators. We will consider recruiting more annotators to have an even more solid human evaluation. Though we think evaluation under the setting is already practical since an acceptable Kappa is observed.
>
> [1] Zero-Resource Knowledge-Grounded Dialogue Generation
>
> [2] Low-Resource Knowledge-Grounded Dialogue Generation
>
>
>
> Q3: The paper does not report Bleu, METEOR or Rouge metrics.
>
> A3: In conversation generation, F1 is one of the essential evaluation metrics, especially in knowledge-grounded conversation tasks. Of course, we could add more evaluation metrics such as BLEU-1, METEOR, and ROUGE-L, and the results are as follows:
>
> | Training Data           | Models    | Wizard Seen |        |         | Wizard Unseen |        |         | CMU_DoG |        |         |
> |-------------------------|-----------|-------------|--------|---------|---------------|--------|---------|---------|--------|---------|
> |                         |           | BLEU-1      | METEOR | ROUGE-L | BLEU-1        | METEOR | ROUGE-L | BLEU-1  | METEOR | ROUGE-L |
> | Reddit Corpus           | BART      | 0.206       | 0.099  | 0.164   | 0.202         | 0.099  | 0.167   | 0.141   | 0.060  | 0.117   |
> |                         | ZRKGC     | 0.225       | 0.099  | 0.163   | 0.220         | 0.100  | 0.164   | 0.161   | 0.075  | 0.128   |
> |                         | Our Model | 0.235       | 0.095  | 0.168   | 0.232         | 0.095  | 0.169   | 0.166   | 0.069  | 0.131   |
> | Reddit Corpus+10\% data | BART      | 0.203       | 0.103  | 0.174   | 0.199         | 0.103  | 0.175   | 0.141   | 0.067  | 0.122   |
> |                         | ZRKGC     | 0.227       | 0.101  | 0.175   | 0.222         | 0.100  | 0.176   | 0.173   | 0.083  | 0.139   |
> |                         | Our Model | 0.237       | 0.103  | 0.181   | 0.231         | 0.101  | 0.176   | 0.182   | 0.080  | 0.139   |
>
> The results indicate that our model outperforms baselines(i.e., BART and ZRKGC) in terms of BLEU-1 and ROUGE-L and achieves competitive performance in terms of METEOR.
>
>
> Q4: The paper had the opportunity to perform further interesting fine-grained evaluation such as accuracy of predicting segment boundaries and accuracy of predicting segment types to better understand the contribution of the two latent variables.
>
> A4: Thank you for your advice. It isn't easy to construct the gold labels of segment boundary/type, so we use pseudo labels constructed according to Section 2 in the Supplementary Material to calculate accuracy. The evaluation results are as follows:
>
> | Training Data           | Models    | Wizard Seen     |             | Wizard Unseen   |             | CMU_DoG         |             |
> |-------------------------|-----------|-----------------|-------------|-----------------|-------------|-----------------|-------------|
> |                         |           | Acc of Boundary | Acc of Type | Acc of Boundary | Acc of Type | Acc of Boundary | Acc of Type |
> | Reddit Corpus           | Our Model | 0.766           | 0.638       | 0.767           | 0.648       | 0.744           | 0.654       |
> | Reddit Corpus+10\% data | Our Model | 0.759           | 0.668       | 0.761           | 0.671       | 0.746           | 0.934       |
>
> The results indicate that the model can achieve promising results in predicting segment boundary/type. Note that there is an imbalance of segment types in the CMUDoG dataset since a large percent of responses in CMU_DoG have little correlation with the knowledge. These results are consistent with the pklg metric in Figure 3.
>
>
>
>
> Q5: The paper does not report numbers by training the proposed model on the whole Wizard/CMUDoG dataset.
>
> A5: Thanks for your advice. Our motivation is to devise a model that could express in a specific style with little annotated data. We believe we have made it clear in section3.1 Problem Formulation and Motivation, at line 111. Maybe a more explicit declaration in the final version could circumvent unnecessary misunderstanding.
> The F1 score of the proposed model on the whole Wizard/CMUDoG dataset are as follows:
>
> |               | Models     | Wizard Seen | Wizard Unseen | CMU\_DoG |
> |---------------|------------|-------------|---------------|----------|
> | (w/o Reddit)  | KnowledGPT | 22.0        | 20.5          | 13.5     |
> |               | Our Model  | 22.0        | 20.8          | 15.3     |
> | (with Reddit) | KnowledGPT | -           | -             | -        |
> |               | Our Model  | 21.9        | 21.2          | 15.5     |
>
> The evaluation results indicate that our model achieves competitive performance on Wizard Seen with a strong baseline KnowledGPT, which is proposed recently and has the best performance on Wizard. In addition, our model outperforms KnowledGPT on Wizard Unseen and CMUDoG significantly which further prove the effectiveness of our model in knowledge-grounded conversation.
>
>
>
> Q6: It is not clear if the proposed model is faithful to the knowledge provided in the documents/paragraphs.
>
> A6: Thanks for your suggestion, and we would like to add human evaluation to explore this issue.
>
>
>
> Q7: About the significance.
>
> A7: We beg to differ about the significance of the improvement. After being trained on 10\% annotated data, our model is supposed to learn the expression style of the Wizard dataset, which is full of long knowledge-relative declarative sentences as we mentioned in section 4. As the expression style is learned and fixed on a specific dataset like Wizard, the diversity metrics (D-1 and D-2) SHOULD decrease. On the contrary, ZRKGC fails to capture the underlying pattern in expression. A little more detailed analysis about the alter of diversity will be included in section 4 in the final draft.
>
>
> Q8: About limitations and societal impact.
>
> A8: Sincerely thank you for your suggestion! It didn't come to our mind at first. We should have paid attention to the negative influence of our work. Abuse of knowledge grounded dialog system could put the audience and the user at the risk of being deceived. We are interested in the controllable generation and the loyalty to the knowledge, which is our future research direction.

---

> > ### Comment · Reviewer_MMWY · 2021-08-17
> > **Further clarification of responses**
> >
> > Thank you for the responses. A few things are still unclear:
> > 1. Thank you for presenting results on additional metrics in A3. It would be nice to also report the other BLEU numbers (especially BLEU-4) similar to the setup in [2]. Would these results be added to the final version of the paper?
> >
> > [2] Low-Resource Knowledge-Grounded Dialogue Generation
> >
> > 2. Thank you for the additional experiments on accuracy of predicting segment boundary and segment types. Would these results be added to the final version of the paper?
> >
> > 3. Thank you for additional result on the whole dataset. Yes, the part about limited annotated data is clear in section 3.1. Again, if the focus of the paper is only on few-shot/zero-shot cases then this should be made more explicit early on in the title, abstract and introduction.
> >
> > 4. Thank you for acknowledging the negative impact of this work. Would you be adding this to the ethics statement? If so, is it possible to present the revised ethics statement in comments?
> >
> > 5. In A6, can you expand on what kind of human evaluation you are planning to explore.

---

> > > ### Author Response · Authors · 2021-08-18
> > > **Respond to Reviewer MMWY**
> > >
> > > Q1: It would be nice to also report the other BLEU numbers (especially BLEU-4) similar to the setup in [2]. Would these results be added to the final version of the paper?
> > > [2] Low-Resource Knowledge-Grounded Dialogue Generation
> > >
> > > A1: Thanks for your kind advice.  For ease of illustration, we only report the results of the BLEU-1 score in the Table (A3) since BLEU-n (n>=3) scores are very close in the dialogue generation task.  We are also glad to provide other BLEU numbers and the results are as follows:
> > >
> > >
> > > | Training Data           | Models    | Wizard Seen |       |       |       | Wizard Unseen |       |       |       | CMU_DoG |       |       |       |
> > > |-------------------------|-----------|-------------|-------|-------|-------|---------------|-------|-------|-------|---------|-------|-------|-------|
> > > |                         |           | B1          | B2    | B3    | B4    | B1            | B2    | B3    | B4    | B1      | B2    | B3    | B4    |
> > > | Reddit Corpus           | BART      | 0.206       | 0.080 | 0.040 | 0.021 | 0.202         | 0.079 | 0.040 | 0.021 | 0.141   | 0.045 | 0.019 | 0.009 |
> > > |                         | ZRKGC     | 0.225       | 0.084 | 0.039 | 0.020 | 0.220         | 0.083 | 0.040 | 0.020 | 0.161   | 0.052 | 0.021 | 0.009 |
> > > |                         | Our Model | 0.235       | 0.087 | 0.040 | 0.020 | 0.232         | 0.086 | 0.040 | 0.021 | 0.166   | 0.052 | 0.021 | 0.010 |
> > > | Reddit Corpus+10\% data | BART      | 0.203       | 0.087 | 0.048 | 0.030 | 0.199         | 0.087 | 0.048 | 0.030 | 0.141   | 0.048 | 0.021 | 0.010 |
> > > |                         | ZRKGC     | 0.227       | 0.097 | 0.053 | 0.032 | 0.222         | 0.093 | 0.050 | 0.030 | 0.173   | 0.060 | 0.025 | 0.010 |
> > > |                         | Our Model | 0.237       | 0.101 | 0.055 | 0.035 | 0.231         | 0.095 | 0.051 | 0.031 | 0.182   | 0.061 | 0.025 | 0.012 |
> > >
> > > We will add these experimental results to the final version of the paper.
> > >
> > >
> > >
> > > Q2: Would these results about additional analyses about segment boundary and segment types be added to the final version of the paper?
> > >
> > > A2: We will definitely add these results to the final version of the paper. Thanks again for these valuable suggestions to help improve our paper.
> > >
> > >
> > > Q3: If the focus of the paper is only on few-shot/zero-shot cases then this should be made more explicit early on in the title, abstract and introduction.
> > >
> > > A3: Thanks for your constructive comments. Our paper focuses on knowledge expression in KGC and we validate our model on both few-shot/zero-shot settings and the regular setting. We will clarify this point in the introduction section of our final vision. Thanks for your valuable suggestions.
> > >
> > >
> > >
> > > Q4: Would you be adding the negative impact to the ethics statement? If so, is it possible to present the revised ethics statement in comments?
> > >
> > > A4: Yes. Here is our updated Broader Impact section which describes the limitations and underlying technical and ethical risks of our work.
> > >
> > > Enabling an open-domain dialogue system to automatically detect and discover the underlying structural pattern of a sentence is of great significance. Capable of handling different expression styles, positive or negative, severe or casual, our work implies that we are stepping closer to the final destination of constructing an artificial intelligent dialogue system that could communicate freely with the human being, which is beyond the wildest dream of most AI and NLP researchers. However, since our model's great capacity, it should be tested for different scenarios before being deposited and applied in real life. A comprehensive survey should be conducted in advance to consider the interest of the direct audience and the developers and all potential groups of stakeholders.
> > >
> > > Besides, knowledge-grounded dialogue systems have the potential to fabricate facts and disseminate rumors and untruths, especially when the source of external background knowledge is not reliable. If the knowledge candidate set is polluted and mixed with fake news, the response generated by the dialogue system is likely to suffer from what we called the ''hallucination'' problem. A necessary and practical method is to strictly control the source of knowledge sentences, such as paragraphs extracted from the wiki, authoritative news sites or authoritative product documents.
> > >
> > > Faithfulness to knowledge is another common issue for a knowledge-grounded dialogue system. Even if provided with correct background knowledge and commonsense, a dialogue system could generate a subjective and chit-chat response without ground in the knowledge set. A blend of personal opinions and objective facts is quite deceptive, and efforts should be made to avoid this. An extension of our proposed model may aid this problem. We could define ''entailed'' as a new content style, the same way as negative and positive styles. In detail, similar to the method mentioned in section 3.3 Learning Details, we can tag entailment/neural/contradiction pseudo labels for knowledge-relevant segments employing a pre-trained natural language inference model and therefore train a set of corresponding adapters to bias the generation process. Another remedy for this problem is to bring in the copy mechanism and equip the decoder with a copy module that could directly copy from the selected knowledge sentences as a segmentation. In all, We believe our model is extendable to handle different scenarios and demands.
> > >
> > >
> > > Q5: In A6, can you expand on what kind of human evaluation you are planning to explore.
> > >
> > > A5: In the Supplementary Material, we have measured the relevance between the generated responses and documents by human evaluation. Of course, this is not enough to verify responses' faithfulness, so we will further explore (1) whether the facts in the response are supported by documents and (2) whether the response contains only objective information. Each annotator will be instructed to assign a score from {0; 1; 2} (representing
> > > ''bad'', ''fair'' and ''good'' respectively) to each response for the two aspects mentioned above. We will add the experiment in our final version.

---

> ### Author Response · Authors · 2021-08-31
> **Thank you**
>
> Thank you for your positive ratings and constructive comments. We will update the final version according to your suggestions.

---

### Official Review · Reviewer_hdNj · 2021-07-16

**Rating:** 6
**Confidence:** 4

**Summary:**

The focus of this research piece is on the introduction of a new model for knowledge-grounded open domain dialogue, that can handle different styles in generation.
The authors identified that in NLG there are two aspects of generation, the knowledge-related, which is based on the prior knowledge and is factual, and the knowledge-irrelevant, which gives the style and the structure of the rest of the generation.
Concretely the authors propose an encoder-decoder model where the three (or more) decoders are available, one based on context, other on prior knowledge and a third or more models which are trained on specific generation style.
A module indicator is trained to choose which model needs to be used.
The model is compared against BART and ZRKGC on  Wizard of Wikipedia and CMU_DoG in terms of F1, Distinct unigrams and bigrams showing improvements over both models training with Reddit and Reddit + 10% of annotated data.
Furthermore, the model is compared against DialoGPT and ECM in terms of sentiment control generation showing a significant difference compared to its competitors

**Limitations And Societal Impact:**

Yes, it seems appropriate

**Main Review:**

Overall a good paper. The motivation is clear, although authors should clarify which aspects of the whole model are in fact novel.
Any NLG paper is meant to have human evaluation since no reliable metric is available. Why didn't the authors include an analysis of that nature as the ZRKGC did?
Why PPL is also missing in automatic evaluation as is included in ZRKGC paper?

Missing references

CTRL: A Conditional Transformer Language Model for Controllable Generation (Keska et al 2019) could be a reasonable competitor on the  sentiment control generation task

**Time Spent Reviewing:**

2

---

> ### Author Response · Authors · 2021-08-10
> **Response to Reviewer hdNj**
>
> Q1: Authors should clarify which aspects of the whole model are in fact novel.
>
> A1: Our novelty lies in (1) the segmentation-based generation component and (2) a variational learning approach to determine the start and the end position of a segment. To our best knowledge, we are the first to study the problem of knowledge-grounded dialog generation from the perspective of segmentation.
>
>
> Q2: Why didn't the authors include human evaluation as the ZRKGC did?
>
> A2: Please refer to Section 8 in the Supplementary Material for more details about human evaluation.
>
>
> Q3: Why PPL is also missing in automatic evaluation as is included in ZRKGC paper?
>
> A3:We did not choose PPL as one of our automatic evaluation metrics at first. The reason is that there is a gap between the training set (Reddit Corpus) and the test set (Wizard and CMU_DoG) in language expression and style. In this condition, PPL is probably not the best metric to evaluate the performance of different models.
> We could further add the PPL metric if necessary. The results are as follows:
>
> | Training Data                     | Models    | Wizard Seen | Wizard Unseen | CMU_DoG |
> |-----------------------------------|-----------|-------------|---------------|---------|
> | Reddit Corpus                     | ZRKGC     | 41.1        | 42.7          | 53.8    |
> |                                   | Our Model | 35.9        | 38.4          | 60.4    |
> | Reddit Corpus+10\% annotated data | ZRKGC     | 29.1        | 31.6          | 38.0    |
> |                                   | Our Model | 28.6        | 30.7          | 40.8    |
>
> The results indicate that our model outperforms ZRKGC on Wizard in both the zero-resource and low-resource settings and is comparable on CMU\_DoG in both settings.
>
>
> Q4: CTRL could be a reasonable competitor on the sentiment control generation task
>
> A4: Thanks for your valuable advice. We further conduct experiments on the CTRL baseline, and the results are as follows:
>
> | Models    | Wizard Seen |      |          |      | Wizard Unseen |      |          |      | CMU_DoG  |      |          |      |
> |-----------|-------------|------|----------|------|---------------|------|----------|------|----------|------|----------|------|
> |           | positive    |      | negative |      | positive      |      | negative |      | positive |      | negative |      |
> |           | F1          | Acc  | F1       | Acc  | F1            | Acc  | F1       | Acc  | F1       | Acc  | F1       | Acc  |
> | CTRL      | 15.3        | 71.9 | 14.9     | 55.3 | 14.9          | 75.0 | 14.6     | 52.3 | 9.3      | 70.2 | 9.4      | 61.7 |
> | Our Model | 19.7        | 70.3 | 19.2     | 70.7 | 19.4          | 73.1 | 19.2     | 69.9 | 12.7     | 74.8 | 12.2     | 68.0 |
>
> The results indicate that our model still outperforms CTRL on sentiment control. In addition, CTRL tends to generate sentences with a positive sentiment, mainly due to the imbalanced distribution of the training data, and our model can alleviate this problem to some extent.

---

> > ### Comment · Reviewer_hdNj · 2021-08-25
> > **OK**
> >
> > Please include the new results as possible,
> > The human evaluation should be part of the main document and should not be left aside on the supplementary materials.
> > Please include the novelties in addition to the contributions as stated in your comments

---

> > > ### Author Response · Authors · 2021-08-29
> > > **Thank you**
> > >
> > > Thanks again for all of your valuable suggestions -- they have substantially contributed to improving the paper, and we will make sure to incorporate them in the final version.

---

### Author Response · Authors · 2021-08-25
**Summary of additional evaluations and clarifications in the author response period**

We appreciate all reviewers for their constructive feedback and comments for the improvement of the paper. We have conducted additional experiments as follows:

1. We compare our model with CTRL (a large-scale pre-trained language model that achieves state-of-the-art performance on controllable generation) on the sentiment control generation task and provide the quantitative results in A4 in response to reviewer hdNj.

2. We evaluate our model and baselines with more automatic metrics, including PPL (please refer to A3 in response to reviewer hdNj) and BLEU, METEOR, ROUGE-L (please refer to A3 in response to reviewer MMWY).

3. We add fine-grained evaluation about the accuracy of predicting segment boundaries and predicting segment types and provide the quantitative results in A4 of response to reviewer MMWY.

4. We further conduct experiments using the whole training corpus of Wizard of Wikipedia and CMU_DoG and compare our model with KnowledGPT, which achieves state-of-the-art performance in knowledge-grounded conversation. The quantitative results are presented in A5 in response to reviewer MMWY.

5. The broader impact session will be revised to contain our proposed model's limitations and negative impact. Possible solutions are also included for the new version of broader impact. Please refer to A4 in the second response to reviewer MMWY.

We hope that these experiments will make our evaluation more reliable and address the reviewers' concerns.

---

### Decision · Program_Chairs · 2021-09-28

**Decision:**

Reject

**Comment:**

This paper explores knowledge-grounded conversation and proposes a new method for better expressing knowledge in conversation by exploiting latent variables encoding either the structure of the response or the style of the content. The reviewers generally think that the approach is intuitive and well-motivated, that it performs well and that the results are important. There were some concerns about the human evaluation in particular, but these were addressed in the author response. I would encourage the authors to be much more civil in future discussions with reviewers and area chairs. As it stands, this paper is borderline, with one reviewer saying the paper is not well-written while two other reviewers argue the exact opposite.

**Consistency Experiment:**

NeurIPS has a long history of experimentation. In 2014, NeurIPS ran an experiment in which 10% of submissions were reviewed by two independent committees to quantify the randomness in the review process. This year, we repeated a variant of this experiment to see how the quality of the review process has changed over time.  This paper was part of the experiment and was therefore assigned to two committees (consisting of reviewers, an Area Chair, and a Senior Area Chair) that reached independent decisions.  If both committees made the same recommendation, this recommendation was followed. If a single committee recommended acceptance, the paper was accepted (with the exception of a few cases in which the other committee identified what we considered a fatal flaw, e.g., an error in a key result).

Both committees reached the same decision: **Reject**

The other committee assigned to the paper recommended **Reject**.  You can find the other set of reviews, along with any follow up discussion with the authors here:
https://openreview.net/forum?id=HQLCFFgK-xl